# An Efficient Ring-Shaped Electromagnetic Thruster

Daniele Funaro [1,*] and Alessandro Chiolerio [2,*]

1 Dipartimento di Scienze Chimiche e Geologiche, Università di Modena e Reggio Emilia, Via Campi 103, 41125 Modena, Italy
2 Istituto Italiano di Tecnologia, Center for Converging Technologies, Soft Bioinspired Robotics, Via Morego 30, 16165 Genova, Italy
* Correspondence: daniele.funaro@unimore.it (D.F.); alessandro.chiolerio@iit.it (A.C.)

**Abstract:** An electromagnetic thruster is proposed and successfully tested. Its design is inspired by theoretical considerations whose qualitative predictions are well matched with the experimental results. The efficiency is higher than any other device so far reported in the literature, producing a directional thrust of approximately $2.7 \times 10^{-6} m$, where $m$ is the mass of the thruster itself, with a nominal power injected of approximately 10 Watts. The prototype has the shape of a ring and is powered by both radio frequency signals and a stationary high voltage. Improvements and generalizations can be easily devised by adjusting the geometry of the device.

**Keywords:** electromagnetic thruster; radio frequency; high voltage; vortex ring

## 1. Introduction

*Cavity thrusters* use radio frequencies (RF) as the only way to generate thrust, thus justifying the name electromagnetic thrusters (EMT). The *EmDrive* [1] is a truncated conical resonant cavity producing a thrust in the direction of the major base when it is fed inside with electromagnetic waves of the proper intensity. The thrust results from the radiation pressure difference due to the asymmetry of the device. Subsequently, other geometries were tested, such as the *Cannae Drive*. Some research groups have actually measured the presence of very little thrust in their tests, but the same experiments conducted by other authors did not confirm the results. Indeed, since the forces were too small, they could have been explained by the interference of thermal effects or by the action of the Earth's magnetic field. Updated information is also available in [2].

Indeed, these devices violate the law of conservation of momentum. One explanation is that an EMT works by transferring momentum to the so-called *quantum vacuum* [3–5]. In fact, the *zero-point energy* is the lowest possible energy state assumed by a quantum mechanical system. Such radiation has an electromagnetic nature and pervades the universe. A typical quantum manifestation is the Casimir effect, where two parallel discharged metal plates kept at a suitable distance are subject to an attractive force [6]. This phenomenon is extremely mild and is usually explained by claiming that the energy of the vacuum outside the plates is larger than the one trapped in the middle, resulting in a gradient of pressure acting on the surfaces.

A similar area of interest concerns asymmetric capacitors, also known as *lifters*. When the device is charged, the two conductors tend to attract each other with a non-zero resultant (Biefeld–Brown effect). In this way, the the entire setup is subject to side acceleration. The so-called *anti-gravity flying machines* are very light capacitors immersed in a dielectric (air, for example). They are charged with a potential difference of the order of tens of kVolts. Due to their asymmetry, the devices start to rise and float freely [7]. The official explanation is based on the fact that there is the production of ionic wind due to a corona-type effect. The asymmetrical movement in the air of these ions would be responsible for the thrust. However, these arguments are not definitively convincing, since some devices seem to

work even when the two capacitor plates are embedded in a solid dielectric material. We refer to the review document [8], which contains a useful list of references and US patent history relating to the subject.

Here, we propose and characterize a new EMT. Its design is prompted by theoretical considerations and takes some inspiration from the aforementioned devices, putting together all of their peculiarities. In particular, the following characteristics are taken into consideration:

- The asymmetry of the device;
- The provision of RF in the "resonant" range of the device;
- The sum of stationary high-voltage electric fields;
- The presence of a dielectric, possibly with a high dielectric constant.

The main argument is based on the following considerations. In the EmDrive, the electromagnetic radiation is injected into the cavity with very little control of what actually happens in there. In our prototype, the wave is specifically driven to form closed patterns. Due to the asymmetry of the paths, the device behaves like an "unbalanced washing machine", spiraling toward a prescribed direction. The assumption is that some sort of "friction" occurs naturally within the electromagnetic vacuum. This should allow momentum to be transferred from the device to the environment. Thus, the asymmetrical dynamic behavior imitates a kind of "swimming" within the omnipresent electromagnetic background. The delay in information transfer due to the finiteness of the speed of light is also a variable to take into consideration. An attempt to explain the phenomenon in the framework of general relativity is given in [9]. Finally, unconventional considerations on mass and gravitation were advanced in [10], Section 2.6 . They might help to solve the puzzle. A theoretical analysis is in any case out of the purposes of this technical report.

We start by introducing particular electromagnetic waves that rotate about an axis. Maxwell's equations in vacuum, involving electric and magnetic fields ($\vec{E}$ and $\vec{B}$, respectively), read in MKS units as follows:

$$\frac{\partial \vec{E}}{\partial t} = c^2 \mathrm{curl}\vec{B} \qquad\qquad \frac{\partial \vec{B}}{\partial t} = -\mathrm{curl}\vec{E} \tag{1}$$

with $c$ denoting the speed of light. The Ampère's law is here implemented without current sources. To the system of Equation (1), we add the following divergence free conditions:

$$\mathrm{div}\vec{E} = 0 \qquad\qquad \mathrm{div}\vec{B} = 0 \tag{2}$$

Peculiar solutions of the whole set of equations, circulating in rounded cavities, are available in [10–13]. For an infinitely long cylinder, the exact expressions can be computed in terms of the classical Bessel's functions. The magnetic field is distributed along the axis of the cylinder, whereas the electric field assumes a dynamic distribution that simulates a rotation around the same axis. The simplest displacement is shown in Figure 1. In the figure, $\vec{B}$ is orthogonal to the page and periodically swings up and down. The figure rotates rigidly at a uniform angular velocity. In this way, the electric field does not remain orthogonal to the direction of motion (in fact, it also shows a longitudinal component) and the entire electromagnetic wave does not travel at a constant speed $c$, equal to that of light. This may seem atypical. On the other hand, it corresponds to what can be directly recovered from solving the set of equations in (1)–(2).

There are infinite such solutions. In the one shown in Figure 1, the magnetic field vanishes (for any time) at the boundary of the cylinder. At the same boundary, the electric field is tangent to the boundary and oscillates according to a function such as $\sin(vt - \phi)$, where $v$ is the peripheral speed of propagation and $\phi$ is the angle. In this scenario, the magnetic and electric fields are coupled via Faraday's law of induction. They both display closed lines. As for $\vec{E}$, these can be clearly recognized in Figure 1, whereas, for $\vec{B}$, the lines are straight and parallel to the axis of the cylinder. In this way, they close at infinity.

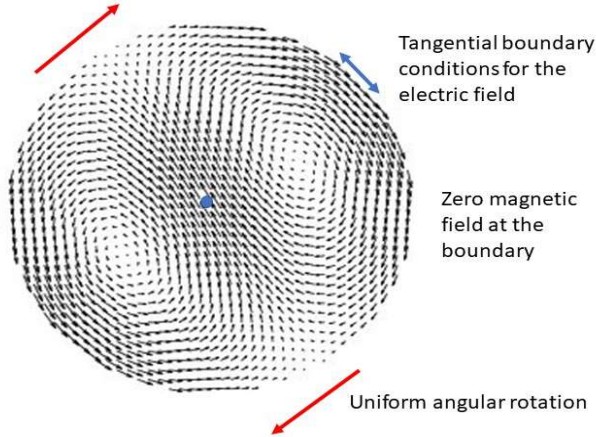

**Figure 1.** Electric field displacement inside the section of the cylinder at a prescribed time. The image rotates clockwise with constant angular speed. This particular solution of Maxwell's equations was constructed with the help of Bessel's functions. The radial component is zero at the boundary, so the field is tangent to the cylinder there. The magnetic field is orthogonal to the page and is also zero at the boundary.

In practice, we should be able to achieve a similar evolution through a winding consisting of a conductive wire (solenoid) positioned around a long dielectric cylinder of fixed diameter. Another rectilinear conductive wire is placed along the axis (i.e., the one passing through the center of the disk in Figure 1) and is connected to ground. The outer wire is then powered by alternating current. The signal is applied to one extreme, whereas the other is kept at a floating potential. This setting guarantees the rotation of the peripheral signal around the axis, which consequently induces a peculiar dynamic field distribution inside the cylinder. The right frequency to be applied depends on various factors, such as the composition of the cylinder, its diameter and the conductivity constant of the outer wire. The resonance of the turns of the outer winding is obtained when the magnetic field generated swings back and forth inside the cylinder in synchrony.

From the case of the cylinder, one can easily pass to that of a ring (not necessarily with a circular section) [14]. Here, the exact solutions of (1)–(2) are not available, but they can be calculated numerically [15]. The evolution of the electric field recalls that of fluid dynamic vortex rings [16]. Now, the magnetic field circulates within the body along closed lines.

By suitably coupling Maxwell's equations with Euler's equation for non-viscous fluids, one can obtain:

$$\frac{\partial \vec{E}}{\partial t} = c^2 \text{curl}\vec{B} - \rho\vec{V} \qquad \frac{\partial \vec{B}}{\partial t} = -\text{curl}\vec{E} \qquad \text{div}\vec{B} = 0 \qquad (3)$$

$$\rho\left(\mu^{-1}\frac{D\vec{V}}{Dt} + \vec{E} + \vec{V}\times\vec{B}\right) = -\epsilon_0^{-1}\vec{\nabla}p \qquad (4)$$

with $\rho = \text{div}\vec{E}$. These modeling equations were firstly introduced in [11]. Thus, we refer to that publication for clarifications. The first equation is the Ampère law, where $\vec{V}$ is a velocity field that describes the evolution of the electromagnetic information (not necessarily consisting of real massive charges, such as electrons). Moreover, the term $D\vec{V}/Dt$ is the substantial derivative, $\epsilon_0$ is the dielectric constant in vacuum and $p$ is a potential denoting a pressure density per unit of surface. Differently from fluid dynamics, $p$ can also take negative values. Under the action of $\vec{\nabla}p$, a surface tends to shift in the direction of lower pressure. In addition, note that the term $\vec{E} + \vec{V}\times\vec{B}$ recalls the Lorentz's force. Finally, the constant $\mu$ is dimensionally equivalent to Coulomb/Kg. An estimate of $\mu$ under a very special circumstance was provided in [10], appendix H. If we set $\rho = 0$, we return to Equations (1) in a vacuum. Thus, the modeling Equations (3)–(4) extend Maxwell ones.

Exact rotating solutions on a cylinder of infinite length can be computed for $\rho = $ constant. This is achieved by linearly combining the solutions for $\rho = 0$ and the stationary one corresponding to a radial electric field (with respect to the axis), as occurs within a dielectric. Such a condition is important for developing a pressure $p$, which is the sum of a steady component and an oscillating one showing zero average over a period of time. The situation is far more complex in the case of a ring, though presents similar characteristics.

In a real experiment, the dielectric constant of the body (cylinder, ring or other more complicated structures) must be taken into account. This has a meaning in determining the displacement of fields. Indeed, we expect that the angular speed of rotation (and the corresponding resonance frequency) varies with dielectric properties. To know what happens inside the body, it is necessary at this level to rely on numerical computations. It is not easy to understand what the balance between the terms in the Equation (4) is. This can vary depending on the orientation of $\vec{E}$. In addition, some quantities grow linearly with the magnitude of $\vec{E}$, whereas others are quadratic. We avoid in this preliminary phase to conduct a more accurate analysis.

For domains that exhibit symmetry (such as the cylinder with a circular section), the instantaneous integral of the pressure gradient is zero, so we do not expect neat forces acting on the body. If we want to generate a non-vanishing resultant (i.e., the integral extended to the whole domain of the pressure gradient), it is necessary to work with an asymmetrical body. We can draw some conclusions considering what happens on the surface, though such an analysis should be performed on the whole body. We also want this resultant to be different from zero when averaged over a period of time. The practical example that we have in mind is shown in Figure 2.

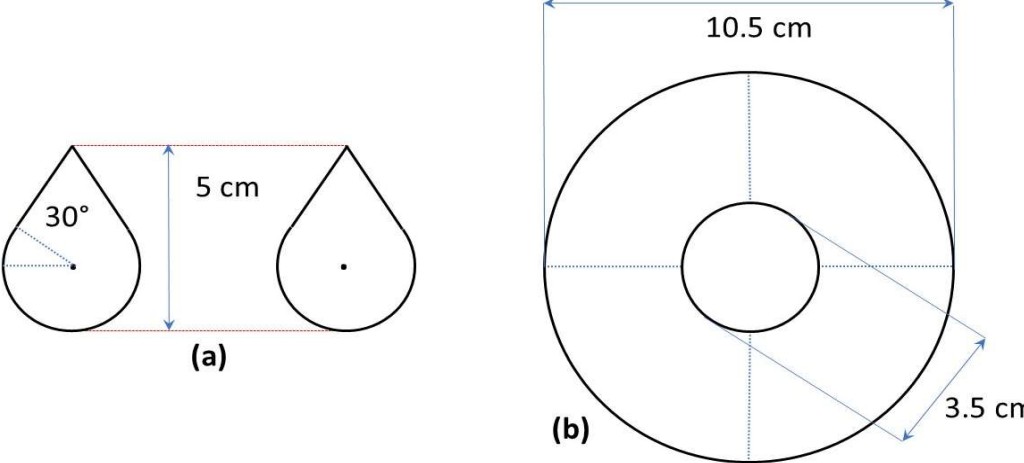

**Figure 2.** Possible configuration of a ring with asymmetric section (section view (**a**); top view (**b**)).

The parameterized elements are:

- The size of the whole device;
- The ratio between the radius of the internal and external diameters;
- The profile of the section;
- The composition of the dielectric.

All of these elements affect the resonance properties of the ring following the application of the RF signal. The limitations imposed on the manufacturing of a real device do not appear to be highly restrictive, however. Hence, there are ample margins for generalizations and improvements. The shape of the ring section should still be the primary concern.

A rough analysis conducted on the surface of the ring of Figure 2 leads us to Figure 3. For a rotating wave, a strong acceleration is produced at the top of the section, where the signal suddenly changes direction. If the section is well designed, we can conclude that there is a neat directional force acting on the body, and the verse of the resultant does not change with time. Note that the ring is not just asymmetrical because the top is

different from the bottom, but also because a descending signal along the internal side (the one corresponding to the hole) and going up along the outer side follows different electromagnetic patterns (i.e., the pitch between the wires is greater on the outside than the pitch between the wires passing through the inside hole).

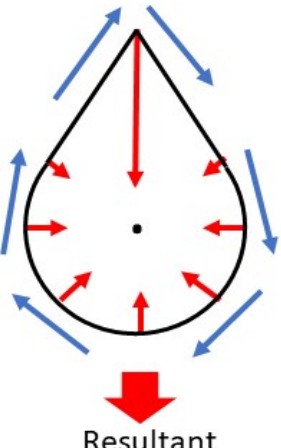

**Figure 3.** Qualitative distribution of the velocity $\vec{V}$ (blue arrows) and acceleration $D\vec{V}/Dt$ (red arrows) along the boundary section of the ring (compare with Figure 2). The section has been designed in order to enforce a sharp variation in $\vec{V}$ on top. This locally results in high values of $D\vec{V}/Dt$ (red arrow pointing down on top), which are not compensated by the integral extended to the remaining portion of the boundary. The final resultant turns out to be different from zero.

If we are to preserve the action–reaction principle, we can argue that, in addition to the internal electromagnetic field, there is an external one, generated by the fact that the device itself acts as an emitting antenna. Thus, part of the energy is asymmetrically radiated outwards. We do not believe, however, that the amount of energy emitted can quantitatively explain the thrust based on radiation pressure theory.

## 2. Materials and Methods

We built the ring according to the following procedure. By using fused deposition modeling (FDM) 3D printing, we created two hollow half-rings as shown in Figure 4, made in PLA, supplied from MakerBot$^{TM}$. The dielectric constant of PLA (and of any other polymer) is strongly influenced by both temperature and frequency and, according to literature, in the GHz range, it is around 3 [17]. One of the two parts supports a circular copper wire that will be connected to ground. The proportions are approximately those shown in Figure 2, and the outer diameter is approximately 10.5 cm. Both parts were filled with a commercial double-component epoxy resin (Henkel, relative dielectric constant approximately 2.5) and then glued together.

Next, as in a toroidal solenoid, the ring was wound by two sets of windings. According to Figure 5, they form two independent circuits. These are interlaced and wound following the same chirality. Here, the parameters that can vary are:

- The number of turns in each winding;
- The width of the wires;
- The metal used for the wires, and therefore its electrical conductivity;
- The chirality.

In our case, there are 65 turns for each wire, made with commercial annealed copper in insulating sheath. The final result is shown in Figure 6.

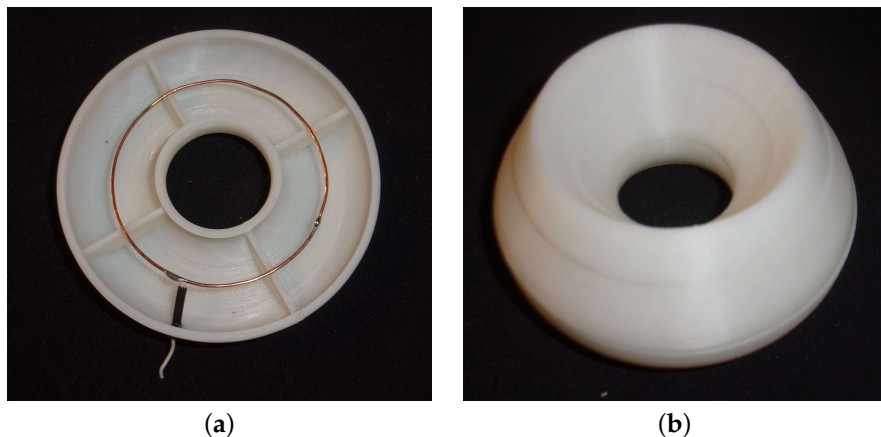

(**a**)　　　　　　　　　　　　　　　　　　　　(**b**)

**Figure 4.** Printed support of the ring (picture **a**), relative to the bottom part. The internal copper loop, to be connected to ground through a small radial conductor, is visible. In (**b**), we can see the two parts assembled. The maximum diameter of the device is approximately 10.5 cm and the proportions are those shown in Figure 2.

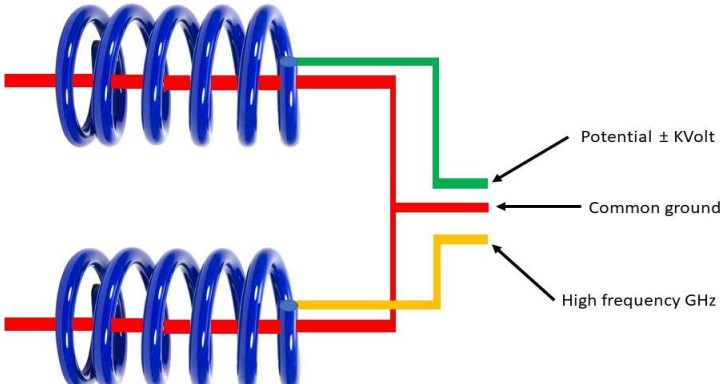

**Figure 5.** Electric scheme.

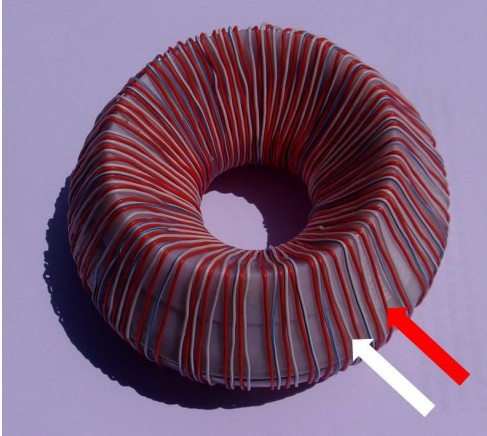

**Figure 6.** Final appearance of the ring. There are two independent coils. The one indicated by the red arrow is supplied with an alternate radio-frequency signal. The one indicated by the white arrow is connected to the high-voltage generator. The common ground is represented by the copper wire running inside the ring as shown in Figure 4a. The corresponding electric diagram is visible in Figure 5.

The endpoint of one of the two wires is connected to the RF signal source, the mass being connected to the central wire within the ring. The other final point is left unconnected. The second wire is connected to a high-voltage generator with the aim to create a

capacitor having the other pole connected to the center grounded wire. This allows for the development of an internal stationary electric field that would provide a constant value of $\rho$ depending on the dielectric characteristics of the material that makes up the ring.

## 3. Results

To measure the thrust generated, we used a mechanical scale (nonius). Several attempts were made to use electronic tools, as also described in Section 4, until we realized that the interference between the RF, high-voltage electric fields and the measuring device would severely affect the experiment. Therefore, we opted for a purely mechanical tool and used a Mettler B5C1000 Laboratory Scale Analytical Balance Machine that can support masses of up to 1 Kg and can appreciate differences of up to 100 μg using an optical vernier. Together with a plastic stand, our device has a total mass of approximately 370 g and is shown in Figure 7.

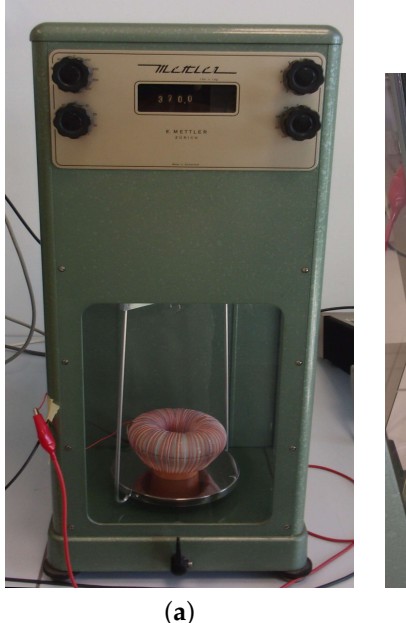 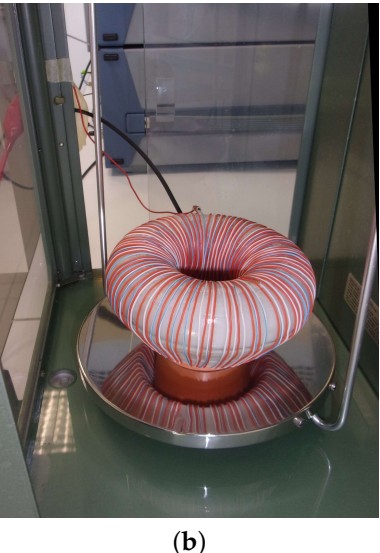

(**a**)　　　　　　　　　　　　　　　　　(**b**)

**Figure 7.** The device on the plate of the balance machine (general view (**a**); enlarged view (**b**)). The orientation is the one referred to in Table 1 as DOWN. The black coaxial cable carries the RF signal. The red cable is connected to the high-voltage generator.

**Table 1.** Absolute mass change (positive = gain, negative = loss) as a function of the electromagnetic fields submitted to the device. The RF frequency is expressed in MHz, the power in dBm, the DC bias in volts and the mass and their standard deviations in $\mu g$. The label UP says that the device is oriented as in Figure 2, whereas DOWN indicates the opposite orientation. We recall that the device has a mass of approximately 370 g.

| Frequency [MHz] | Power [dBm] | DC Bias [V] | Device Orientation | Mass Variation [μg] |
|---|---|---|---|---|
| 913 | 5 | 0 | UP | $400 \pm 50$ |
| 913 | 5 | 99.95 | UP | $400 \pm 100$ |
| 913 | 5 | 250 | UP | $450 \pm 100$ |
| 913 | 5 | 1000 | UP | $400 \pm 100$ |
| 913 | 5 | 0 | DOWN | $-700 \pm 200$ |
| 913 | 5 | 1000 | DOWN | $-900 \pm 100$ |
| 940 | 5 | 0 | DOWN | $-600 \pm 150$ |
| 940 | 12 | 0 | DOWN | $-800 \pm 100$ |

As a frequency generator, we used a Rohde & Schwarz SWM02, which provides frequencies from 0.01 up to 18 GHz. The maximum output power is 12 dBm, equal to

15.85 mW, while most of the experiments were conducted at 5 dBm, equal to 3.16 mW. The signal is driven by a 10 W (nominal power) amplifier powered by 12 volts. The high-voltage DC generator was a Keithley 2410 1100V SourceMeter. The current fed through the circuit is negligible, except for the transient one, necessary to charge the capacitor made up of the dielectric material that separates the inner ring and one of the two outer wires (compare with the scheme in Figure 5 ). This produces a value of $\rho \neq 0$ inside the ring, which can be positive or negative depending on the polarity of the applied voltage.

At frequencies between 913–940 MHz, the balance shows consistent and systematic variations in the weight of the EMT. This range is compatible with the size of the device, according to information circulating at speeds of the order of that of light. In particular, the EMT was tested in different conditions, varying the RF frequency and power, the DC bias and the orientation of the ring with respect to the vertical axis (up-down or down-up, in relation to the scheme of Figure 2). The frequencies were changed manually with steps of 1 MHz. The response profile looks Gaussian, showing a width of approximately 30–40 MHz.

Figure 8 shows examples of the results obtained in the experiments that we performed by keeping the value of DC bias fixed and repeatedly switching on/off the RF power approximately every 10 s. A mass variation of slightly less than 1 mg is observed for a maximum RF power not exceeding 10 watts. The effect of DC bias was also evaluated (Figure 9), while the radio frequency was maintained.

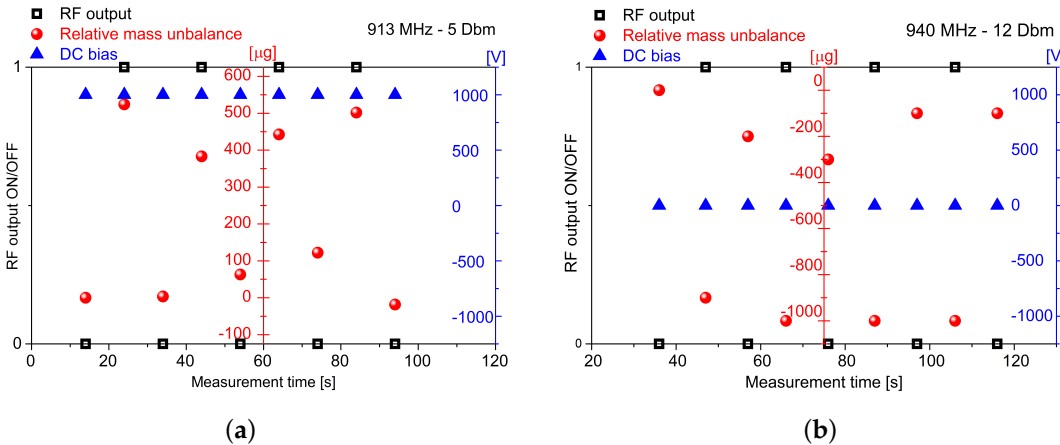

(**a**)  (**b**)

**Figure 8.** Outcomes relative to two typical experiments performed while applying the RF field and a DC bias. The RF source is switched on and off periodically approximately every 10 seconds of acquisition (black open squares: '1' represents the device on, '0' represents the device off), at 913 MHz at the power of 5 Dbm (panel **a**), and 940 MHz at the power of 12 Dbm (panel **b**). The DC bias is set at a fixed value (blue triangles). The balance readings are converted into a relative variation scale by taking the initial value (red circles) as zero and eventually removing the small linear drift that can sometimes be found, presumably due to heating/cooling phenomena during the day. The measure in (**a**) was performed positioning the EMT as shown in Figure 2, whereas the one in (**b**) was performed by reversing the orientation.

The figures of merit are: a mass variation of approximately $2.7 \times 10^{-6} m$, where *m* is the mass of the EMT, and a maximum thrust of approximately 9.8 µN. Despite the low power injected, the efficiency turns out to be at least one thousand times higher than the results achieved previously by other authors. The reverse orientation of the ring causes a change in the sign of the force exerted, and therefore a loss of mass, instead of a mass gain.

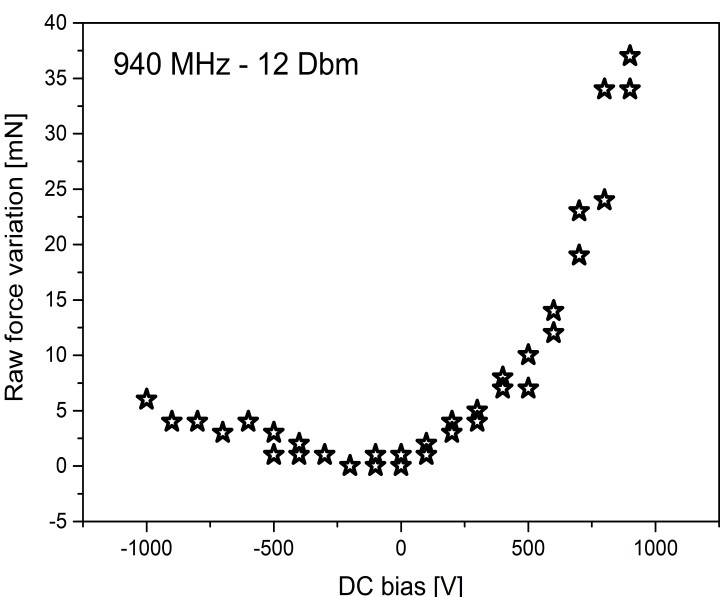

**Figure 9.** Effect of the high voltage DC bias on the force variations while the RF field is kept on at the frequency and power indicated in the inset. Here, the device is oriented as in Figure 2. The voltage was varied from −1 kVolt to 1 kVolt and backwards, with steps of 100 volts, completing a full hysteresis loop. The indication of raw force variation is due to the fact that data are untreated to compensate for any eventual thermal drift.

## 4. Discussion

Based on the results of the experiment, the following facts were verified:

- The effect is systematic. It clearly appears and disappears by turning the RF power on and off. There is a slight viscous lag related to the stabilization of the mechanical balance. Linear drift is sometimes observed during the measurements, possibly due to temperature variations in the laboratory and instrumentation.
- The result strictly depends on the frequency. Reasonably, this is related to the resonance properties of the device. We did not systematically check the entire frequency range, mainly because the amplifier did not support a large band.
- The strength of the outcome is proportional to the applied power. Without the amplifier the effect is hampered.
- The phenomenon is also observed without applying the high-voltage component. We observed a marginal improvement in performance by applying a potential of ±1 kV, as shown in Figure 9.
- Applying high voltage without the high-frequency signal produces no result, as expected.
- The direction of the thrust depends on the verse of the orientation of the device, and points downwards as shown in Figure 3. This is the most important property, which validates the entire experiment.

For completeness, we report here a brief history of the experiments carried out in the past on similar devices. Soon after the publication of the book [11] (based on the preliminary partial version [18]), a rough prototype was built as shown in Figure 10. The electron model introduced in [11], Section 5.3 (and successively re-examined in [10], Appendix H) inspired the realization of such a ring-shaped device, with a wiring formed by 208 coils. The idea is that time-varying electric fields can autonomously create regions where the divergence $\rho$ is different from zero (see, for instance, the explanation given in [19] in the case of a pulsating charge). This activates the pressure term $p$ in (4) with the consequent generation of Newtonian-like forces. The size of the ring (total diameter around 70 cm) was decided depending on the frequency generator available at that time, working at the frequency of approximately 1 GHz. The lack of adequate instrumentation and the absence of an internal

solid dielectric (not essential in principle, but necessary for realistic results) did not allow for an observation of appreciable effects. A possible asymmetry of the device was not taken into consideration. At that time, the purpose was not to gain directional thrust, but to create some kind of very mild gravitational shielding. This aspect was also suggested by further observations (as reported later in [10], p. 58). For practical impediments, this project was abandoned.

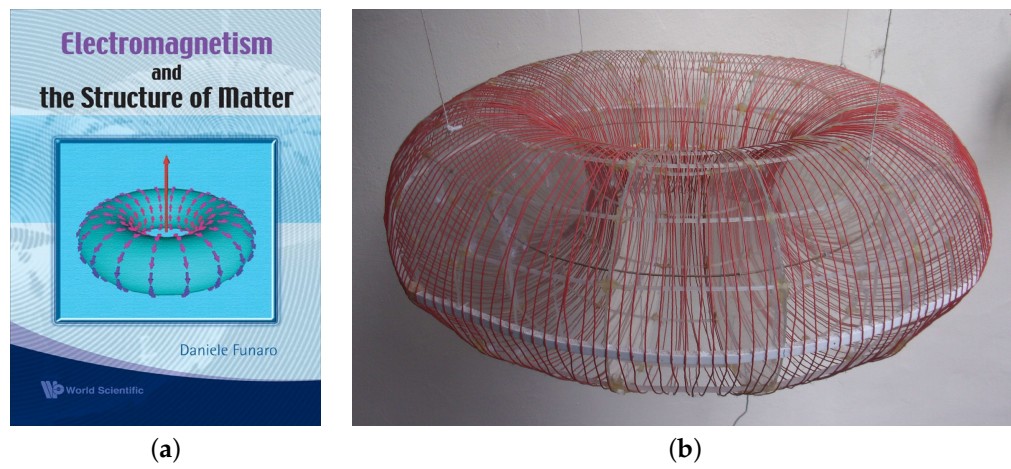

|  |  |
| :---: | :---: |
| (**a**) | (**b**) |

**Figure 10.** The cover of the book (**a**) and an archaic version of the ring (**b**).

Another experiment was attempted in 2014 at the Physics Department of Politecnico di Torino. In this new device shown in Figure 11, an electromagnetic wave is generated by a quarter-wavelength dipole antenna. A copper ring is connected to a high-voltage source (up to 5 kV). The purpose was to make the wave circulate around the ring in closed patterns, thus generating mechanical pressure (see also the computational results in [20]). For some theoretical assumptions, we expected a higher pressure compared to the classic electromagnetic radiation pressure, which is notoriously extremely mild. In light of the experiences described here in Section 3, that idea now seems very naive. The forces were measured through the continuous monitoring of the resonance of a silicon cantilever, amplified by the deflection of a laser beam focused on the reflecting surface of the cantilever itself, and collected by a four-quadrant photodiode, as shown in Figure 11 . When the forces, mediated by the emitting device, act on the volume that also contains the resonant cantilever, a variation in the resonant frequency is expected. The test failed due to an interference between the electromagnetic wave and the solid-state laser instrument that produced artifacts. This is also the reason why, in the experiments of Section 3, we opted for a purely mechanical balance.

The possibility of introducing a dielectric was considered years later. A material with a high dielectric constant at the operational frequencies of the experiment reinforces the density $\rho$ in (4) and slows down the transfer speed of electrodynamical information, allowing for the use of smaller objects within a specific frequency range.

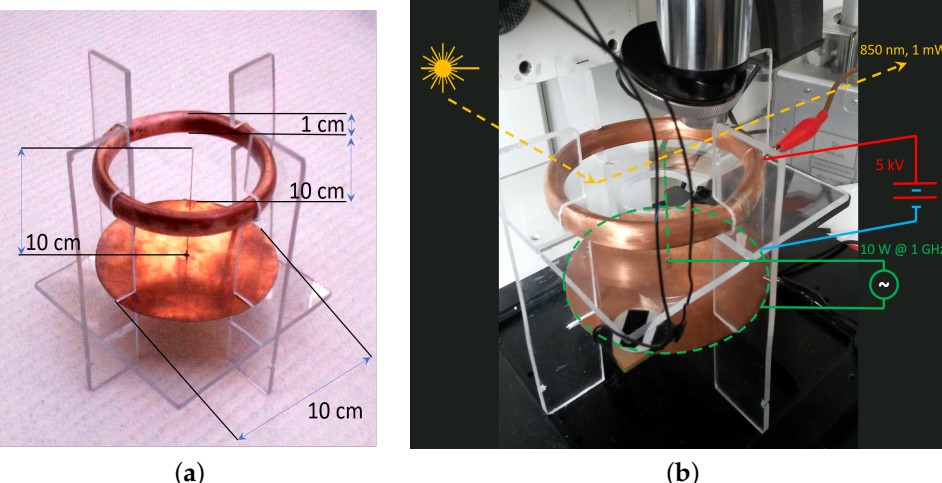

**Figure 11.** The structure of the device tested in 2014 is shown in (**a**), with quotes reported in cm. The same device under test is shown in (**b**). The labelling describes how the measurement was conducted. The acrylate transparent stand was used to keep the ring and RF antenna in position on the one side, and to provide a stable horizontal surface where the cantilever is positioned, to explore the pressure effects. The yellow arrows represent the laser beam (1 mW @ 850 nm) directed to the cantilever and reflected toward the four-quadrant photodetector. Red and blue lines represent the high-voltage (5 kV) connection scheme (the ring on the one side, the copper dish on the other), whereas green lines represent the RF (10 W @ 1 GHz) connection scheme (the central copper wire and the dish). A silicon cantilevered chip was positioned at the reflection point of the laser beam, centered using the microscope objective visible at the top of Figure 11b . The piezoelectric actuator that triggers the oscillation of the silicon cantilever was located on the front of the picture, fixed to the horizontal acrylate plate. The power was supplied via two black wires vertically aligned and running through the experiment volume from the microscope objective to the plate. The signal to power the piezodrive, locked to the four-quadrant sensor output, had an amplitude of 10 V and a frequency between 20 Hz and 50 kHz.

## 5. Conclusions

An EMT apparatus was successfully designed and tested. It is based on a ring geometry and responds to the combined effect of high-frequency signals and stationary high voltage. The impact on the mass $m$ of the body is equivalent to a variation of approximately $\pm 2.7 \times 10^{-6} m$, corresponding, in our case, at an applied force of 9.8 μN for a nominal applied power of 10 W. More thrust is expected by injecting additional power. There is a large amoount of room for improvement while playing on the geometry and wiring of the prototype. In the apparent violation of the principle of conservation of momentum, the device can be used in all applications where, in the absence of moving parts, magnets or external fueling, it is necessary to impart acceleration to a frame by means of an electromagnetic input. This phenomenon can be of fundamental importance in many engineering applications, such as in the development of new space propulsion units. Our invention does not require the use of any solid/liquid propellant, thus offering huge savings and greater reliability, not needing it for pump systems, valves, nozzles, etc. It only requires electricity, like any other instrument or calculator onboard. For orbital positioning, where smaller forces/impulses are needed, the EMT is perfect. Further applications may be air pumping/conditioning, water pumping, etc.

**Author Contributions:** D.F.: conceptualization, theoretical formalization, design, assembly of the components, arrangement and supervision of experiments, first draft of the paper and editing of the final version. A.C.: technical discussion, performing experiments and data analysis, public relations, contacts, editing of the final version of the paper. All authors have read and agreed to the published version of the manuscript.

**Funding:** This research received no external funding.

**Institutional Review Board Statement:** Not applicable.

**Data Availability Statement:** Data are available upon reasonable request to the authors.

**Acknowledgments:** The successful test discussed in this paper is the result of the help offered by various people of Dipartimento di Scienze Chimiche e Geologiche and the Dipartimento di Ingegneria 'Enzo Ferrari' of the Università di Modena e Reggio Emilia. In particular, we would like to thank Andrea Marchetti and Claudio Fontanesi for providing the laboratory instrumentation, Francesco Gherardini and Enrico Dalpadulo for the 3D printing of the ring, Lorenzo Tassi for the filling with epoxy resin and Moreno Maini for some preliminary electric measures on the device. Concerning the previous experiments mentioned in Section 4, we would like to thank Marco Crepaldi (Istituto Italiano di Tecnologia, Genova) and Carlo Ricciardi (Dipartimento Scienza Applicata e Tecnologia, Politecnico di Torino) for their kind support. Thanks also go to Alfredo Currado for some clarifying theoretical discussions.

**Conflicts of Interest:** The authors declare no conflict of interest.

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
