# Peer review of "An Efficient Ring-Shaped Electromagnetic Thruster"

_inventions, doi:10.3390/inventions8020051_

Round 1

Reviewer 1 Report

The manuscript entitled ‘An Efficient Ring-Shaped Electromagnetic Thruster' submitted for review has several issues that need to be corrected and/or significantly improved:

The equations marked with the No1 (1) and (2) are not adequately enumerated and described in the text.

The Figure 1 should be described and explained in the text above the appearance of the figure. In addition, the Figure 1 should be equipped with appropriate legend and description.

Figure 3 is not clearly described.

The dimensions and all component need to be described in the Figure 4.

In my opinion the Wikipedia do not need to be cited.

All physical quantities should be indicated in Figure 11.

The subtitle '7. A bit of history' is not appropriate.

The material properties need to be clearly explained and discussed.

Author Response

1. Response to Reviewer 1
The manuscript entitled ‘An Efficient Ring-Shaped Electromagnetic Thruster’ submitted for review has several issues that need to be corrected and/or significantly improved:
The equations marked with the No1 (1) and (2) are not adequately enumerated and described in the text.
We have split and differently enumerated equations (1) and (2) in section 2. More description has been provided and the corrections are reported in red.
The Figure 1 should be described and explained in the text above the appearance of the figure. In addition, the Figure 1 should be equipped with appropriate legend and description.
Figure 1 has been described both in the text and in the caption. We moved the figure.
However, regarding the final positioning within the text the last choice is left to the editorial office that will take care of the final formatting of the paper.
Figure 3 is not clearly described.
Figure 3 has been modified to make it clearer. The caption is much more lengthy and contains the necessary information.
The dimensions and all component need to be described in the Figure 4.
In Figure 4 we modified the caption to better describe the components, as requested by the Reviewer. In addition, in order to provide all necessary information about the size of the device, we inserted quotes in Figure 2.
In my opinion the Wikipedia do not need to be cited.
The reference to Wikipedia has been removed.
All physical quantities should be indicated in Figure 11.
We have added to both panels of Figure 11 the relevant physical quantities: geometry size, laser beam features, high voltage and RF supply details. The caption has been adapted
accordingly.
The subtitle ’7. A bit of history’ is not appropriate.
The title of section 7 has been changed to: "History of past experiments".
The material properties need to be clearly explained and discussed.
The materials used for manufacturing the EMT and their properties have been specified at the beginning of Section 5.

Reviewer 2 Report

Dear Authors,

Please find attached a PDF file with some comments.

Author Response

1. Response to Reviewer 2
Dear Authors, Please find comments as follows:
Line 1: abbreviation EMT is not used in the abstract. Please skip it.
As requested, we did not use the abbreviation EMT in the abstract.
Lines 308-335: please format the references with the MDPI format.
We have reformatted the references according to the MDPI style.
Line 56: please indicate the unit system you have used.
We have indicated the system used, i.e. MKS.

Reviewer 3 Report

The following minor revisions are recommended to the authors:

1) Refer that the left one of the four equations in (1), relative to the Ampere's law is applied when there are no current sources;

2) Refer that the equation div E_ = 0 in (1), is valid because there are no current and electrical charge sources, and the electric field is created by induction according to the Faraday's Law expressed by dB_/dt=-curl E_, from the closed alternating magnetic flux density B_ lines. Because of this reason the induced electric field in the disk presents closed lines and equation div E_=0 verifies in this particular scenario;

3) Figure 1 should be placed after the paragraph between lines 58 and 68 where it is referred for the first time in the text;

4) Figures 2, 4, 7, 8, 10, and 11 should be divided in part (a) and part (b), detailing was is shown in each of the drawings, photos, or graphics;

5) In the right photo of Figure 7, new Figure 7(b), insert two labels with arrows indicating which are the coils supplied by DC bias potencial and RF signals;

6) In the two graphics of Figure 8 insert units in the vertical axes, and increase a little more fonts in legends and axes;

7) Insert units in the header of Table 1;

8) Whenever is written "(see Fig. X)" replace by ",as shown in Fig. X,";

9) Figure 11 should appear after the paragraph between lines 257 and 271, where it is referred for the first time in the text.  In line 258 replace "In this new device (Figure 11)," by "In this new device shown in Figure 11,";

10) The interest in this article could be improved if the authors identify possible applications for Ring-Shaped EMT.

Author Response

1. Response to Reviewer 3
The following minor revisions are recommended to the authors:
1) Refer that the left one of the four equations in (1), relative to the Ampere’s law is applied when there are no current sources;
We have specified in section 2. that the Ampère law is implemented without current sources. These corrections and the successive ones are reported in red.
2) Refer that the equation div⃗E = 0 in (1) is valid because there are no currents and electrical charge sources, and the electric field is created by induction according to the Faraday’s Law expressed by d⃗B/dt = −curl⃗E, from the closed alternating magnetic flux density ⃗B lines. Because of this reason the induced electric field in the disk presents closed lines and equation div⃗E = 0 verifies in this particular scenario;
We have specified in Section 2 that the equation div⃗E = 0 is valid because there are no currents and electrical charge sources. Moreover, we noticed that this property is actually verified by the field displayed in Figure 1.
3) Figure 1 should be placed after the paragraph between lines 58 and 68 where it is referred for the first time in the text;
We moved the figure. However, regarding the final positioning within the text the last choice is left to the editorial office that will take care of the final formatting of the paper.
4) Figures 2, 4, 7, 8, 10, and 11 should be divided in part (a) and part (b), detailing was is shown in each of the drawings, photos, or graphics;
As suggested, figures 2, 4, 7, 8, 10, and 11 have been divided in part (a) and part (b).
5) In the right photo of Figure 7, new Figure 7(b), insert two labels with arrows
indicating which are the coils supplied by DC bias potential and RF signals;
Because of the lack of space we preferred to add to Figure 6 the arrows indicating which are the coils supplied by DC bias potential and RF signals. The caption has been modified accordingly.
6) In the two graphics of Figure 8 insert units in the vertical axes, and increase a little more fonts in legends and axes;
In Figure 8 we have inserted units in the vertical axes and increased by 30% the fonts in legends and axes.
7) Insert units in the header of Table 1;
Units in the header of Table 1 have been added.
8) Whenever is written "(see Fig. X)" replace by ",as shown in Fig. X,";
Throughout the text, whenever was written "(see Fig. X)", we have replaced ",as shown in Fig. X,"
9) Figure 11 should appear after the paragraph between lines 257 and 271, where it is referred for the first time in the text. In line 258 replace "In this new device (Figure 11)," by "In this new device shown in Figure 11,";
We moved the figure. However, as also specified above, regarding the final positioning within the text the last choice is left to the editorial office that will take care of the final formatting of the paper. Moreover, as requested, "In this new device (Figure 11)," has been changed with "In this new device shown in Figure 11,"

10) The interest in this article could be improved if the authors identify possible applications for Ring-Shaped EMT.
Possible additional applications of the EMT have been detailed at the end of the conclusions section.

Round 2

Reviewer 1 Report

The manuscript has been improved and thus it can be considered for acceptance.